# Pyrolysis and Oxidative Thermal Decomposition Investigations of Tennis Ball Rubber Wastes through Kinetic and Thermodynamic Evaluations

**DOI:** 10.3390/ma16062328

**Published:** 2023-03-14

**Authors:** Hai-Bo Wan, Zhen Huang

**Affiliations:** 1Department of Physical Education, Tianjin University of Commerce, Tianjin 300134, China; 2Department of Packaging Engineering, Tianjin University of Commerce, Tianjin 300134, China

**Keywords:** tennis ball rubber wastes, thermal degradation, thermogravimetric analysis, isoconversional kinetics

## Abstract

Thermal decomposition of tennis ball rubber (TBR) wastes in nitrogen and air has been studied through thermogravimetric analysis. The samples were thermally decomposed from room temperature to 950 K at heating rates of 3 to 20 K/min with a purging flow of 30 cm^3^/min. The degradation features and specific temperatures for two purging gases are thus compared according to the nonisothermal results. Kinetic analyses of two thermal decomposition processes have been isoconversionally performed using differential or integral methods. The activation energy as a function of mass conversion has been thus obtained over the entire decomposition range, varying from 116.7 to 723.3 kJ/mol for pyrolysis and 98.2 to 383.6 kJ/mol for oxidative thermal decomposition. The iterative Flynn–Wall–Ozawa method combined with the linear compensation effect relationship has been proposed for determining the pre-exponential factor and reaction mechanism function, resulting in chemical order reaction models of *f*(*α*) = (1 − *α*)^5.7^ and *f*(*α*) = (1 − *α*)^5.8^ for describing pyrolysis and the oxidative thermal degradation of TBR wastes, respectively. With these kinetic parameters, very satisfactory matching against experimental data has been obtained for both gases. Additionally, the thermodynamic parameters, such as the changes of entropy, enthalpy and Gibbs free energy, over the whole thermal degradation processes have also been evaluated.

## 1. Introduction

Currently, tennis has gained global popularization since it is of great assistance to maintain people’s mental wellness and physical fitness. Therefore, millions of people are attracted to this sports activity every year, leading to the rapid and vigorous development of the tennis industry around the world. Along with this increased interest, a significant amount of tennis balls has been used and then discarded as one kind of municipal solid waste after use. As such, it will be a great impetus to promote the tennis economy if the tennis wastes, such as disused tennis rubber balls, can be properly disposed. Until now, thermochemical conversion technologies, such as pyrolysis, combustion and gasification, are widely accepted as economically sustainable processes since polymer-based or biomass wastes can be environmentally friendly by being converted into flammable fuels or industrial chemicals [1,2]. On the other hand, the conventional methods of disposing into landfills or incineration are no longer preferred since they irreversibly produce hazardous gaseous pollutants from incineration [3,4] or irreparably consume valuable solid resources in the landfill [5].

Therefore, pyrolysis and combustion have been extensively investigated for the purpose of sustainable disposal of various solid wastes [1,2,6]. Recently, we have made exploratory efforts on how to properly dispose of tennis wastes via inert pyrolysis, and oxidative decomposition has been thus conducted [7,8], where thermal characteristics and kinetic analysis have been reported in detail. In the present work, tennis ball rubber (TBR) waste has been considered for the thermal disposal purpose and then studied by means of thermogravimetric analysis (TGA). Thereafter, non-isothermal TGA data under multi-heating rates can be experimentally obtained for the elaboration of the thermal decomposition features and kinetics of TBR wastes so that the information required for designing a thermochemical conversion reactor may thus be readily available. For achieving such a purpose, many attempts have been made to study the pyrolysis kinetics of rubber wastes [9,10,11,12,13] and these works have detailed the kinetic process in terms of the activation energy *E*_k_, pre-exponential factor ln*A* and reaction model function *f*(α). Since industrial rubber products are generally made up of over ten ingredients, the thermal decomposition process of rubber wastes may involve very complicated reactions, usually including two or more stages [9,10,11,12,13]. For this reason, the *E*_k_ and ln*A* are seen to span in a very wide range of 3~283 kJ/mol and 7.56 × 10^2^~1.39 × 10^19^ min^−1^ [11], while for *f*(α), the reaction order model is preferably used [10]. Furthermore, a number of researchers have also considered the compositions and yields of thermally conversed products along with kinetic investigations [11,14,15,16,17]. For instances, Miranda et al. [14] studied pyrolysis of rubber tyre wastes mainly composed of natural rubber (NR), styrene–butadiene rubber (SBR) and cis-1,4-polybutadiene rubber (BR), and found that by using gas chromatography (GC), the liquid fraction includes alkanes, alkenes and aromatic compounds, whilst Yu et al. [15] reported that the main gaseous products are CH_4_, H_2_ and C_2_H_4_ when studying pyrolysis of tire rubber/Fe_2_O_3_.

In the present work, the pyrolysis and oxidative thermal decomposition behaviors and kinetics of TBR wastes have been studied via non-isothermal TGA measurements for deeply understanding the effect of the purging atmospheres on the degradation processes by comparing the thermal features in terms of specific temperatures, the heat-resistance index [18] and the devolatilization index [19]. Furthermore, kinetic modeling using experimental data has been performed to determine three kinetic parameters of *E*_k_, ln*A* and *f*(α) so as to gain valuable insights into a given process and provide a rigorous prediction of the thermal processes. According to the ICTAC Kinetics Committee recommendations [20,21,22], model-free and model-fitting methodologies can be appropriately attempted for performing kinetic analysis. To obtain the *E*_k_ values, a number of model-free methods involving no assumption of any reaction model have been detailed, such as the differential Friedman (FD) method [23], the integral distributed activation energy model (DAEM) method [24] and the integral Flynn–Wall–Ozawa method [25]. It may be noted that the FD method seems to be theoretically the most accurate, but the integral methods involve different approximations and, thus, inevitably lead to certain inaccuracies. For this reason, an iterative-FWO method [25] or a nonlinear regression procedure [20] has been proposed for the integral methods to improve the accuracy of the *E*_k_ values. Secondly, the model-fitting method can be performed by fitting different reaction models against TGA data to result in the most proper kinetic model function. For depicting thermal degradation processes, the nth-order reaction models, diffusion-controlling models, phase boundary reaction models and nucleation and nuclei growth reaction models have widely been attempted according to the physico-chemical or physico-geometrical features of reactants [22]. In the case of polymer degradations, the first-order or nth-order models seem to be the foremost selection in many works, although they may have produced inconsistent results [22]. Finally, the estimation of the pre-exponential factor in terms of ln*A* can be made by preferably making full use of the compensation effect (CE) relationship. Prof. Vyazovkin recently [26] highlighted that by using all ln*A* and *E*_k_ pairs, correct or not, the CE approach is highly accurate for both kinetic single-stage and multi-stage pyrolysis processes, and the model-free ln*A* values can be accurately yielded once the accurate *E*_k_ values are available.

In present work, the FD method, DAEM method and it-FWO method have been used to calculate the conversion-dependent *E*_k_ values, while the compensation effect relationship [13,19] combined with the it-FWO method [25] has been attempted here for the determination of ln*A* and *f*(*α*) so as to reveal the complex thermal degradation reaction. With these kinetic parameters, rebuilding the mass conversion curves has been performed, consequently resulting in very satisfactory predications against the experimental results. In addition, the thermodynamic parameters during the thermal degradation processes have also been quantified by following the transition state theory [27]. The results of present work will offer experimental and kinetic information necessary for designing any thermo-conversion reactor to thermally treating TBR wastes.

## 2. Materials and Analysis Methods

### 2.1. Materials

The tennis ball rubber (TBR) wastes were collected locally in Tianjin, China but were made in the Philippines by the brand ^®^Slazenger. After removing the outside wool fabric cover, the inner rubber samples were cut into pieces and pulverized with a mill, yielding powders with a mesh size of less than 100 μm. Prior to conducting the decomposition experiments, the powders were heated at 105 °C for 2 h and thereafter kept in a desiccator.

### 2.2. TGA Analysis

A thermogravimetric analyzer (DTG-60, Shimadzu, Japan) was employed for the thermal degradation measurements following a constant heating rate mode. For each run, a powdery sample with a mass of approximately 4.5 mg was placed into the sample crucible and then heated from room temperature to 1000 K under 3, 5, 10, 15 and 20 K/min, with the product of the sample mass and heating rate clearly satisfying the ICTAC Kinetics Committee recommendations [21]. The purging gas used was inert N_2_ or oxidative air with a flow rate of 30 cm^3^/min. The experimental data were automatically recorded and the first derivatives of the TGA (DrTGA) data were subsequently produced with the analyzer software.

### 2.3. Pyrolysis Characteristic Parameters

For the estimation of the thermochemical performances of TBR wastes in inert N_2_ or oxidative air atmosphere, two characterization parameters were used and they are the heat-resistance index (*HRI*) [18] and the comprehensive performance index (*CPI*) [7,19], respectively. The mathematical expressions for the two parameters are given below:(1)$$HRI=0.49\times [T_{5}+0.6(T_{30}-T_{5})]$$
(2)$$CPI=\frac{{DTG}_{p}\cdot {DTG}_{mean}}{T_{i}T_{p}∆T}$$
where *T*_5_ and *T*_30_ represent the absolute temperatures at the mass conversion equaling to 5% or 30%, respectively. *DTG*_p_ and *DTG*_mean_ are the maximum mass loss rate (min^−1^) and the average mass loss rate (min^−1^) of interest, respectively. *T_p_* is the temperature at *DTG*_p_, while *T_i_* represents the temperature at which the thermal decomposition of the TBR waste begins. Finally, ∆*T*_0.5_ is defined as the temperature range corresponding to *DTG*/*DTG*_max_ = 0.5. A higher *CPI* value usually indicates a better thermochemical performance.

### 2.4. Kinetic Analysis

The kinetic analysis of the thermochemical process may be of great help to better understand the effect of the temperature and time on the thermal decomposition features. Usually, the rate of any thermal decomposition process can be written in terms of the decomposition temperature (*T*, K), decomposition time (*t*, min) and the extent of decomposition conversion (α, dimensionless), as in the following:(3)$$\alpha =\frac{w_{i}-w_{t}}{w_{i}-w_{f}}$$
(4)$$\frac{d\alpha }{dt}=A\cdot \exp\left(-\frac{E_{k}}{RT}\right)f(\alpha )$$
(5)$$k(T)=A\cdot \exp\left(-\frac{E_{k}}{RT}\right)$$
where *w*_i_, *w*_t_ and *w_f_* are defined to be the initial, instantaneous, and final sample masses, respectively, and these data can be directly abstracted from the experimental TGA results. In the meantime, *R* is the universal gas constant (8.314 J/(K·mol)), and *A* and *E_k_* are the pre-exponential factor (min^−1^) and the apparent activation energy (kJ/mol), respectively. It may be noticed that *k*(*T*) is the rate constant of the decomposition reaction and *f*(α) is the reaction mechanism function to control the thermal decomposition.

Under non-isothermally experimental conditions, the most popular temperature program is to keep the heating rate constant with a parameter *β* defined as *β* = d*T*/d*t* and then the following expression can be easily deduced:(6)$$\frac{da}{dT}=\frac{A}{\beta }\cdot \exp\left(-\frac{E_{k}}{RT}\right)\cdot f(a)$$

Alternatively,
(7)$$g\left(\alpha \right)=\int_{0}^{\alpha }\frac{d\alpha }{f(\alpha )}=\frac{A}{\beta }\int_{0}^{T}\exp\left(-\frac{E_{k}}{RT}\right)dT$$
where *g*(*α*) also stands for the reaction mechanism function, but in the integral form. It should be highlighted here that the temperature-integral function in the right-hand side of Equation (7) has no analytical solution and it might be solved numerically or approximated to different extents. As for the reaction mechanism functions available either in differential *f*(*a*) or in integral *g*(*a*), some literature can be referred to for the details [7,8,28].

#### 2.4.1. Friedman Differential (FD) Method

The FD method [23] is a model-free isoconversional method to estimate the α-dependent *E*_k_ values and it can be directly derived from Equation (2) without any approximation. For this reason, the FD method has been widely applied for any thermochemical degradation processes. Equation (2) can be conveniently written in logarithmic expression:(8)$$\ln\left(\frac{d\alpha }{dt}\right)=\ln\left[A\cdot f\left(\alpha \right)\right]-\frac{E_{k}}{RT}$$

Accordingly, at any particular value of α, *f*(α) will be a constant and plotting ln(d*α*/d*t*) versus the reciprocal temperature of 1/*T* will lead to a straight line for each α. The activation energy *E*_k_ may be resulted from the slope of –*E*_k_/*R* over the entire conversion range.

#### 2.4.2. Distributed Activation Energy Model (DAEM) Method

Unlike the FD method, the DAEM method is an integral method and originally developed to kinetically describe the decomposition process of various materials [11,24,29]. In this method, an infinite number of irreversible first-order reactions are assumed to constitute in parallel the whole thermochemical decomposition processes. Accordingly, the following expression may be used to directly obtain the activation energy and pre-exponential factor:(9)$$\ln\left(\frac{\beta }{T^{2}}\right)=\ln\left(\frac{AR}{E_{k}}\right)+0.6075-\frac{E_{k}}{RT}$$

Thus, the plot of ln(*β*/*T*^2^) against 1/*T* will give a straight line. The *E*_k_ and ln*A* values for a given *α* value can be estimated from the slope and intercept of the resultant line, respectively. 

#### 2.4.3. Iterative Flynn–Wall–Ozawa (It-FWO) Method

Based on the conventional FWO method or other analogies, Gao et al. [25] proposed a new iterative procedure to augment the accuracy of calculating kinetic parameters. Accordingly, the it-FWO method can be simply written as the following expression:(10)$$\ln\left(\frac{\beta }{H(x)}\right)=ln\frac{0.0048AE_{k}}{Rg(\alpha )}-1.0516\frac{E_{k}}{RT}$$
(11)$$H(x)=\frac{Ω(x)}{0.0048exp(-1.0516x)}$$
(12)$$Ω\left(x\right)=\frac{exp(-x)}{x^{2}}\cdot \frac{x^{4}+18x^{3}+86x^{2}+96x}{x^{4}+20x^{3}+120x^{2}+240x+120}$$
where *x* = *E*_k_/*RT*. Clearly, by taking *H*(*x*) = 1, the linear plots of ln[*β*/*H*(*x*)] against 1/*T* for all *α* values are identical to those of the conventional FWO method, leading to *α*-dependent *E*_k_ values from the slopes (−1.0516*E*_k_/*R*) of these lines. To make the it-FWO method work, a few steps to conduct the iterative calculations may be taken [25]: (i) considering *H*(*x*) = 1, the activation energy values of *E_k_*_1_ will simply be produced. (ii) Using *E_k_*_1_ to calculate *Ω*(*x*) and *H*(*x*), and thereafter plotting the straight lines of ln[*β*/*H*(*x*)]~1/*T* will result in a new set of *E_k_*_2_ values from the line slopes. (iii) Replacing *E_k_*_1_ with *E_k_*_2_, the iteration as above is repeated to obtain another new set of *E*_k_ values, and such iteration will stop until |*E*_kn_ − *E_k_*_n−1_| < 0.1 kJ/mol at the iteration where *n* is satisfied. (iv) Taking the final set of *E_k_*_n_ values as the exact activation energy values, the *α*-dependent *E_k_* values are then obtained over the entire conversion range. 

Usually, the it-FWO method is taken as a model-free method to identify the dependence of *E*_k_ on *α*, but it is rarely used for the determination of the reaction model and the pre-exponential factor. Therefore, it has been modified for such a purpose with the aid of the compensation effect (ln*A* = *a*·*E*_k_ + *b*). The compensation effect has been widely accepted to give accurate evaluations of the pre-exponential factors [26]. With the *E_k_* values thus obtained above, the intercept of each it-FWO plot at the last iteration *n*, *I_n_*, is taken as the following:(13)$$I_{n}=ln\frac{0.0048AE_{kn}}{R\cdot g(\alpha )}$$

Applying the compensation effect will lead to Equation (14):(14)$$Y_{n}=I_{n}-\ln\left[\frac{0.0048AE_{kn}}{g\left(\alpha \right)\cdot R}\right]=lnA=a\cdot E_{kn}+b$$

Thereafter, the model-fitting efforts are made by plotting *Y_n_* against *E_kn_* over 0.05 < *α <* 0.95 for all *g*(*α*) models [7,8,28]. For simplicity, one global reaction model is assumed herein for the entire kinetic degradation process. The fitting performance for a specific kinetic model *g*(*a*) can be judged by evaluating the linear correlation coefficient *R*^2^ and then the best appropriate model is taken as the one with the *R*^2^ value closest to 1.0. Once the *g*(*α*) function is determined to be the most probable model, the compensation parameters *a* and *b* can be obtained from the slope and the intercept of the correspondent fitting line, respectively. Subsequently, the lnA can be computed with *a*, *b* and *E*_kn_ as the function of *α*. 

### 2.5. Thermodynamic Analysis

The thermodynamic parameters, such as enthalpy (Δ*H*), entropy (Δ*S*) and Gibbs free energy (Δ*G*), were calculated over the whole pyrolysis and oxidative decomposition process of the TBR waste. For such a purpose, the transition-state theory [27] could be attempted, thereby leading to the following expressions [7,8]: (15)$$A=\frac{ek_{B}T}{h_{P}}\exp(\frac{\Delta S}{R})$$
Δ*H* = *E_k_* − *RT*
(16)
Δ*G* = Δ*H* − *T*Δ*S*
(17)
where *e* represents the Neper number (2.7183), while *k*_B_ and *h_P_* stand for the Boltzmann constant (1.381 × 10^−23^ J/K) and the Plank constant (6.626 × 10^−34^ J/s), respectively. 

## 3. Results and Discussion

Figure 1 presents the TGA and DrTGA results of the TBR waste samples obtained in air and N_2_ at 10 K/min. It may be readily observed from the DrTGA curves that for the TBR waste, its degradation process seems to consist of four consecutive zones in inert N_2_ or oxidative air, indicative of rather complex decomposition reactions. Similar observations have also been reported in the literature [11,15]. From the results shown in Figure 1, it can be seen that the main decomposition of TBR wastes appears to take place in Zone I and Zone II for the case of the oxygen-rich condition and only in Zone I for the case of the oxygen-free condition. Their correspondent mass losses are 83.01 and 77.85%, respectively. Furthermore, four different zones may be estimated according to the variation of the DTG results and they are 417.1~696.5, 696.5~749.1, 749.1~830.8 and 830.8~903.5 K for the air case, and 401.8~589.3, 589.3~749.2, 749.2~791.4 and 791.4~995.9 K for the N_2_ case, respectively. It may be noted that such estimations are very rough and the observation of multiple zones may be due to the presences of many components in the rubber wastes [11]. The mass percentage of the final residual for either purging gas is essentially the same, around 35%, as shown in Figure 2. Figure 2 and Figure 3 present the TGA and DrTGA results of the TBR waste in air and N_2_ under different heating rates, respectively. As can be observed, the TGA and DTG curves shift to higher temperature ranges as the heating rate increases, and this shift is usually considered to be the result of temperature hysteresis [7,8]. In the meantime, the *T*_p_ and *DrTGA*_p_ values are also seen to increase with the heating rate and Table 1 can be referred to for more detail.

Table 1 and Table 2 present some characteristic parameters for the thermal decomposition of TBR wastes in air and N_2_. As can be seen from Table 1, the *HRI* value goes up slightly as *β* increases, indicating the little influence of the heating rate on the thermal resistance. A comparison between the two purging gases shows that these *HRI* values are generally comparable to each other for each heating rate, consistent with similar *T*_5_ and *T*_30_ values for both cases. Table 2 lists a few feature parameters of *T*_i_, *T*_p_, *DrTGA*_p_, *DrTGA*_mean_ and Δ*T* directly abstracted from the TGA and DrTGA results for four thermal decomposition stages at 10 K/min. According to Equation (2), the *CPI* value for each zone can be readily calculated, and, interestingly, the *CPI* value in Zone II is the highest for both gases, even though Zone I for the air case seems to exhibit the largest mass loss. The mass loss values of four Zones are 64.32, 18.69, 11.96 and 5.03% for air, and 8.35, 77.85, 3.17 and 10.63% for N_2_ gas, respectively. As such, the *CPI* values can be simply estimated by using the mass addition method [7] and the results are given in Table 2, as well. By comparing the *CPI* values, it may be plausibly deduced that nitrogen is better than air to make TBR wastes thermally degradable. 

### 3.1. Kinetics Analysis of Thermal Degradation 

#### 3.1.1. Determination of E_k_ with the FD Method

With the FD method, the plots of ln(d*α*/d*t*) against the reciprocal temperature are presented in Figure 4 for the thermal decomposition of TBR waste in air and N_2_. Thereafter, the *E_k_* values are estimated from the slopes of the linear Arrhenius plots and these data are graphically shown in Figure 5 over the entire conversion range. Also shown in Figure 5 are the correspondent linear correlation coefficient *R*^2^ results for all *α* values, and it can be seen that the *R*^2^ is almost equal to 1.00 for most *α* < 0.7 values, revealing a very good linear dependence of ln(*β*/*T*^1.92^) on 1/*T*. However, for *α* > 0.7 values, the *R*^2^ is relatively far away from 1.0, indicative of rather poor linear dependence. 

Clearly, the *E_k_* values seen in Figure 5 heavily depend on α for both cases of oxidative air or inert N_2_, whereas they vary in the range of 98.2~383.6 kJ/mol and 131.2~723.3 kJ/mol when α spans with 0.05 < α < 0.95 and the averaged *E_a_* values are 268.3 and 306.0 kJ/mol over the whole conversional range. In the case of oxidative air, the *E_k_* increases from 173.0 to 356.0 kJ/mol as α goes up from 0.15 to 0.50 and it rapidly drops to 98.2 kJ/mol when α progresses further to 0.85. On the other hand, the *E_a_* values for pyrolysis are seen to increase from 131.2 to 307.2 as α increases from 0.05 to 0.70, and then soar up to 723.3 kJ/mol at α = 0.85, followed by a sharp decline to 223.8 kJ/mol at α = 0.95. These results tend to suggest that the pyrolysis or oxidative decompositions apparently have involved complex chemical reactions of multiple stages. The differences in *E*_k_ as observed above for the air and N_2_ cases indicate that the purging atmosphere may have strongly influenced the thermochemical decomposition features of TBR wastes.

#### 3.1.2. Determination of E_k_ with the DAEM Method

The DAEM method was also studied for isoconversionally analyzing the thermal decomposition kinetics of TBR wastes in air and N_2_. According to Equation (9), the linear Arrhenius plots of ln(*β*/*T*^2^) against 1000/*T* are demonstrated in Figure 6. Other than the FD method, a much better linear relationship between ln(*β*/*T*^2^) and 1/*T* can be seen over the whole conversion range of 0.05 < *α* < 0.95, agreeing well with the *R*^2^ values shown in Figure 5. For each *α* value, the *E_k_* value is calculated from its correspondent Arrhenius line slope and the *E_k_* results thus calculated over 0.05 < *α* < are also graphically presented in Figure 5. Compared to the *E_k_* results in Figure 5, it may be seen that the *E_k_* value resulted from the DAEM method has shown a trend similar to that obtained using the FD method. In the case of oxidative degradation, the *E_k_* spans from 135.1 to 375.7 kJ/mol over the range of 0.05 < *α* < 0.95. On the other hand, the *E_k_* values are seen to alter from 116.7 to 602.9 kJ/mol for N_2_ pyrolysis. Overall, the averaged *E_k_* values are 272.2 and 249.6 kJ mol^−1^ for the thermal decomposition of TBR wastes in air or N_2_, respectively. These *E*_k_ results, varying widely with the progress of mass conversion, are somewhat consistent with those reported for pure rubbers, such NR, BR and SBR, in the literature [30,31,32]. For example, Perejόn et al. [30] separated complex pyrolysis of natural rubber into two decomposition stages and the averaged *E_k_* values were found to be 205.2 and 318.9 kJ/mol for the first and second stages, respectively. Danon et al. [31] reported that the averaged *E_k_* values are 215 and 434 kJ/mol for two pyrolysis stages of NR, 409 and 244 kJ/mol for two pyrolysis stages of BR and 305, 235 and 80 kJ/mol for three pyrolysis stages of SBR, respectively. Similarly, Conesa and Marcilla [32] reported that the averaged *E_k_* values are 212, 290 and 45 kJ/mol for three pyrolysis stages of BR, and 267 and 385 kJ/mol for two pyrolysis stages of SBR, respectively. All these works tend to suggest that thermal decomposition processes are rather complicated and more care should be taken when performing kinetic analysis. 

#### 3.1.3. Determination of E_k_ with the It-FWO Method

Following the it-FWO method, the *E_k_* values are iteratively estimated and the iteration will stop, provided that the condition of *E*_*k*, *j* + 1_ − *E*_*k*, *j*_ < 0.1 kJ/mol is satisfied for each *α*. Shown in Figure 7 are the it-FWO results satisfactorily meeting the convergence requirements. The linearity between ln[*β*/*H*(*x*)] and the reciprocal temperature *T* are very good over 0.05 < *α* < 0.95, as also reflected by the *R*^2^ values in Figure 5. The *E_k_* values are then calculated from the slopes of these Arrhenius plots and the results are also given in Figure 5 for both air and N_2_ cases. As can be clearly seen, the *E_k_* data for all *α* values are within 135.7~375.9 kJ/mol in air and 117.0~603.1 kJ/mol in N_2_, respectively, very close to those from the DAEM method. From this regard, it may be deduced that these two integral methods are generally equivalent to estimate the *α*-dependent *E_k_*. When compared to those from the FD method, the *E*_k_ values are comparable. 

#### 3.1.4. Compensation Effect Considerations

Apart from the *E*_k_ data obtained as above, the other kinetic parameters ln*A* and *f* (*α*) have also been considered. If following the DAEM method, the global first-order reaction mechanism of *f*(*α*) = 1 – *α* is taken. Accordingly, the ln*A* is then calculated using Equation (9) and they are shown as a linear relationship with the *E*_a_ values. The ln*A* values thus obtained vary within 20.1~65.0 min^–1^ for air and 21.3~99.2 min^–1^ for N_2_, respectively. It may be noted that the linear relationship between the *E*_k_ and ln*A*, usually called the kinetic compensation effect, is relatively good, as revealed by the *R*^2^ values shown in Figure 8. 

With the kinetic triplet data thus available, the DAEM method is then used to correlate the experimental data for TBR wastes and Figure 9 graphically presents the correlated results against the experimental data. Clearly, it can be seen that most data points are scattered closely around the diagonal line rather than condensed on the line, indicative of relatively large deviations from the DAEM method as compared to experimental values. Thus, a model-fitting method is usually preferred for obtaining the best-performing correlations. Here, the modified it-FWO method has been applied to serve this purpose based on the combination of the iterative FWO method with the kinetic compensation parameters (ln*A* = *a·E*_k_ + b).

Based on the activation energies iteratively obtained, the ln*A* value can be evaluated from the intercepts of the linear it-FWO plots by inserting a g(*α*) function into Equation (14). Even though there are four stages for the TBR wastes’ decomposition process, the one global reaction model assumption has been considered for the entire kinetic degradation process. After having scanned all the theoretical models [7,8,23], the reaction-order functions are found to achieve better satisfactory fittings, and the linear compensation effect plots from the best models are shown in Figure 8. For the air and N_2_ cases, the best models are the F5.8 and F5.7 functions with the forms of *f*(*α*) = (1 − *α*)^5.8^ and *g*(*α*) = [(1 − *α*)^−4.8^ − 1]/4.8 or *f*(*α*) = (1 − *α*)^5.7^ and *g*(*α*) = [(1 − *α*)^−4.7^ − 1]/4.7, respectively. It may be noted that the reaction order of more than 5.0 is less frequently reported [20], but it can be used to obtain the best fit against the TGA data. That is to say, these values can provide straightforward modeling results against experimental curves, but they may have a clear physical meaning. Using the F5.8 and F5.7 functions, the lnA values thus estimated vary within 23.1~107.5 min^–1^ for N_2_ and 30.5~69.5 min^–1^ for air, respectively. As can be clearly seen from Figure 8, two ln*A*~*E*_k_ plots possess better linearity than those from the DAEM method, as reflected by their correspondent *R*^2^ values of 0.9978 and 0.9934, respectively.

As discussed above, the kinetic parameters from the modified it-FWO method are also attempted to match the experimental results, and Figure 9 presents the matching results against the experimental values for TBR waste in both air and N_2_. Very satisfactorily, for both the air and N_2_ cases, all the data points have almost condensed on the diagonal line, tending to suggest the excellent fittings for the respective decomposition processes of TBR wastes.

### 3.2. Thermodynamic Parameter Calculation

Apart from the kinetic parameters, the thermodynamic parameters (Δ*H*, Δ*G* and Δ*S*) for thermal decomposition of TBR wastes have also been estimated using Equations (15)−(17), where the *E*_k_ and ln*A* values used are from the it-FWO method, and these results are shown in Figure 10. As can be seen from Figure 10a, all the calculated Δ*H* values are positive, indicating that both the inert pyrolysis and oxidative thermal decompositions are endothermic processes. The values of Δ*H* are found within the range of 129.1–369.1 kJ/mol for air and 112.3–596.9 kJ/mol for N_2_, and the average values of ΔH are determined to be 266.7 and 244.3 kJ/mol, respectively. The Δ*H* values for TBR within a conversion fraction of 0.1 < α < 0.7 are higher in air than in nitrogen, possibly related to its high molecular weight. The higher positive Δ*H* values infer that a greater amount of heat is required for thermal decomposition [7,8,28].

Additionally, the Δ*S* and Δ*G* for TBR thermal degradation are also evaluated, and the obtained data are presented in Figure 10b,c, respectively. As seen from Figure 10b, the large Δ*S* values are found to vary from −8.1 to 317.7 J/mol·K for air or from −66.0 to 632.9 J/mol·K for N_2_, respectively. Thermodynamically, Δ*S* is a parameter to quantify the extent of disorder for any specific process of a targeted system. The positive Δ*S* value is readily understandable as a result of thermally cracking highly-ordered macromolecules into small molecules of high freedom. As for the negative Δ*S* value, it indicates relatively slow thermal decomposition reactions and the system is far from the equilibrium state [28]. From Figure 10c, the Δ*G* values for the air or N_2_ cases are found to be 114.7–155.9 kJ/mol and 120.8–159.1 kJ/mol, respectively. The positive values of Δ*G* both confirm that thermal decompositions of TBR wastes in either air or nitrogen are a thermodynamically non-spontaneous process, and this is natural, since the forced thermal decomposition is made through a nonisothermal TGA program. The higher Δ*G* value infers less favorability of a specific reaction, suggesting that a larger amount of heat is required for thermal decomposition.

## 4. Conclusions

In the present study, pyrolysis and oxidative thermal decomposition of tennis ball rubber wastes have been studied non-isothermally. The thermal features are thus compared and the kinetic parameters are calculated for describing pyrolysis and oxidative thermal degradation of TBR wastes. Some conclusions may be drawn, as follows.

The thermo-oxidative decomposition and pyrolysis of TBR wastes involved multiple reaction stages, and the thermal characteristic parameters of *T*_5_, *T*_30_, *T*_p_, *DrTGA*_p_ and *HRI* were found to increase with the heating rate. For simplifying the thermal decomposition processes of TBR wastes, the one global reaction model is taken for conducting a kinetic analysis.

Over the range of 0.05 < α < 0.95, the *E_k_* values for the thermal degradation of TBR wastes are estimated to vary within 98.2~383.6 kJ/mol for air and 116.7 to 723.3 kJ/mol for N_2_, respectively. In the meantime, the ln*A* values were found to alter within 23.14~107.49 min^–1^ for N_2_ and 30.45~69.51 min^–1^ for air, respectively.

Among three model-free methods, both the DAEM and it-FWO methods yielded almost the same *E_k_* values over the entire conversion range, and the *E_a_* values resulted from the differential FD method are generally comparable to those from the other two methods.

By means of integrating the it-FWO method with the linear compensation effect, scanning theoretical reaction models were globally performed accordingly and *g*(*α*) = [(1 − *α*)^−4.8^ − 1]/4.8 and *g*(*α*) = [(1 − *α*)^−4.7^ − 1]/4.7 other than *g*(*α*) = −ln(1 − *α*) were finally found to be the most suitable mechanism function for describing pyrolysis and oxidative degradation of TBR wastes, as verified by the excellent matching against the experimental data.

The thermodynamical parameters of Δ*H*, Δ*G* and Δ*S* are estimated to range within 129.1~369.0 kJ/mol, 114.7~155.9 kJ/mol, and −8.1~317.7 J/mol·K for oxidative degradation, or 112.3~596.9 kJ/mol, 120.8~159.1 kJ/mol and −66.0~632.9 J/mol·K for N_2_ pyrolysis, respectively. These results indicate it is unfavorable to thermally decompose TBR wastes, especially for more energy-required pyrolysis.

## Figures and Tables

**Figure 1 materials-16-02328-f001:**
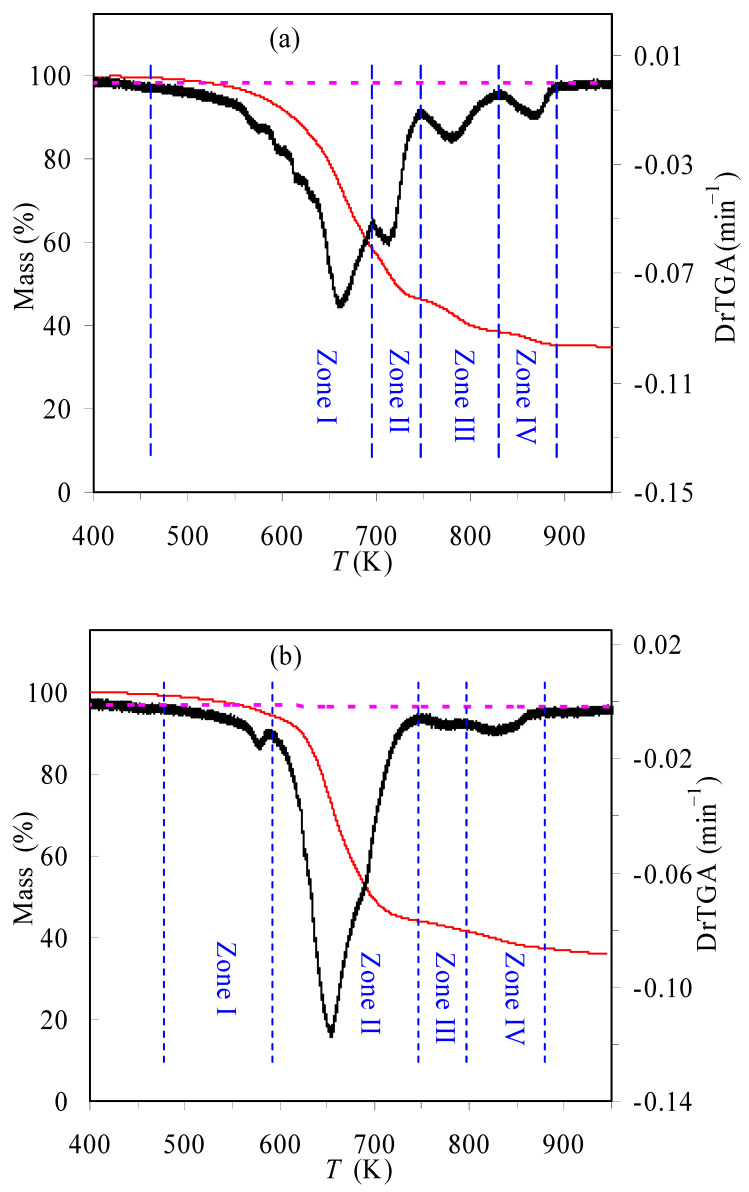
TGA and DrTGA results of TBR wastes in air (**a**) and N_2_ (**b**) at 10 K/min.

**Figure 2 materials-16-02328-f002:**
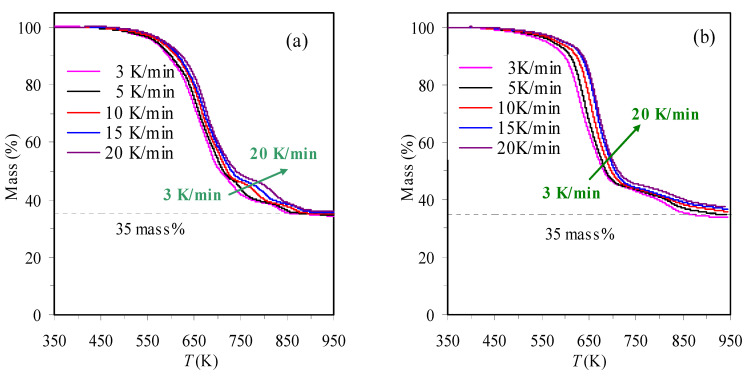
TGA results of TBR wastes in air (**a**) and N_2_ (**b**) determined under different heating rates.

**Figure 3 materials-16-02328-f003:**
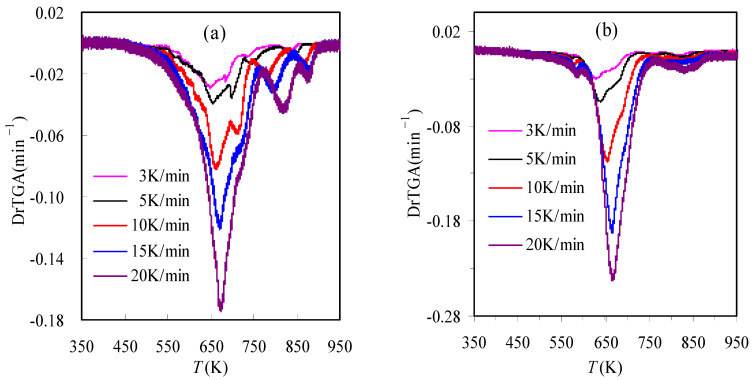
DrTGA results of TBR wastes in air (**a**) and N_2_ (**b**) determined under different heating rates.

**Figure 4 materials-16-02328-f004:**
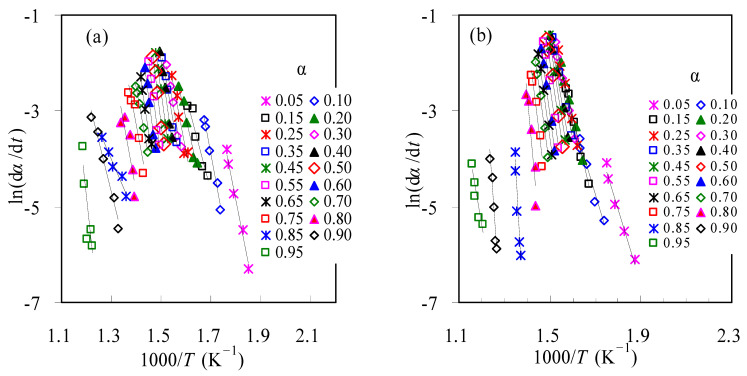
The linear FD plots of ln(d*α*/d*t*)~1000/*T* for thermal decomposition of TBR wastes in air (**a**) and in N_2_ (**b**).

**Figure 5 materials-16-02328-f005:**
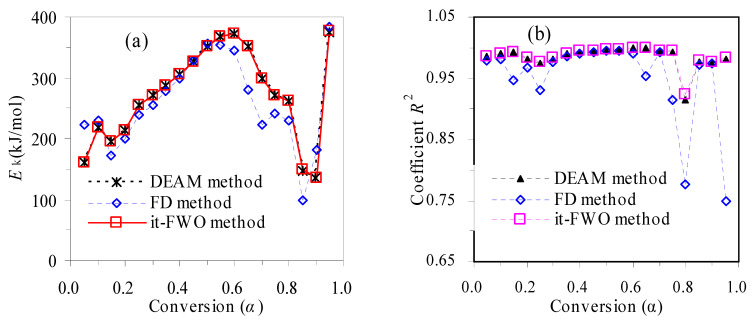
α-dependent *E*_k_ (**a**,**c**) and linear correlation coefficients (**b**,**d**) by different methods for TBR wastes in air (**a**,**b**) and N_2_ (**c**,**d**).

**Figure 6 materials-16-02328-f006:**
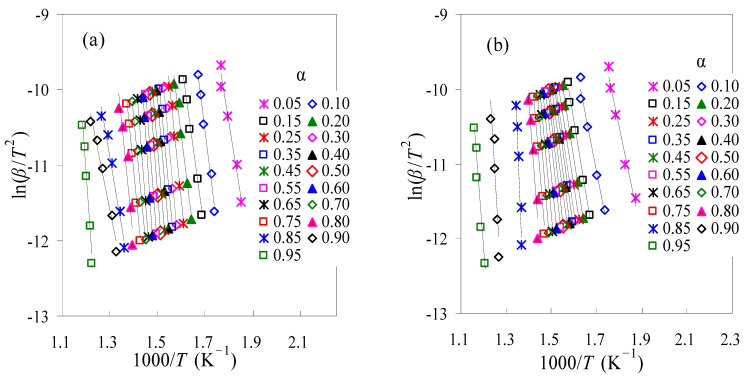
The linear DEAM plots of ln(d*α*/d*t*)~1000/*T* for thermal decomposition of TBR wastes in air (**a**) and in N_2_ (**b**).

**Figure 7 materials-16-02328-f007:**
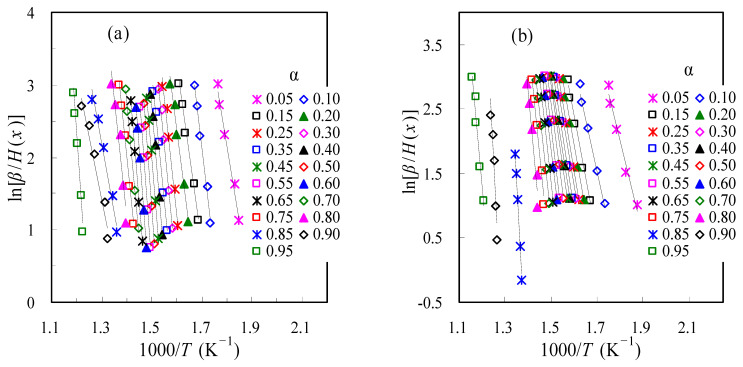
The linear it-FWO plots of ln[*β*/*H*(*x*)]~1000/*T* for thermal decomposition of TBR wastes in air (**a**) and in N_2_ (**b**).

**Figure 8 materials-16-02328-f008:**
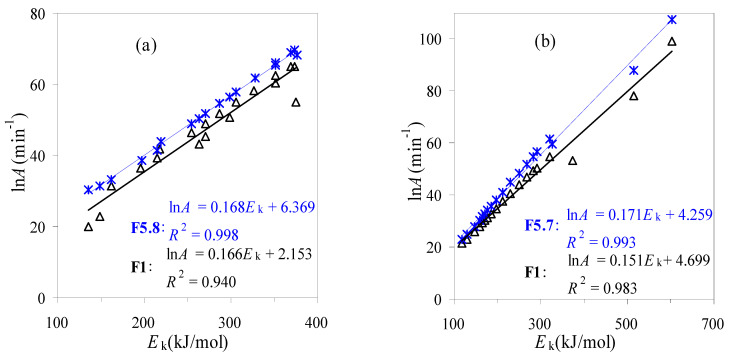
Compensation effect between ln*A* and *E_a_* for thermal decomposition of TBR wastes in air (**a**) and in N_2_ (**b**).

**Figure 9 materials-16-02328-f009:**
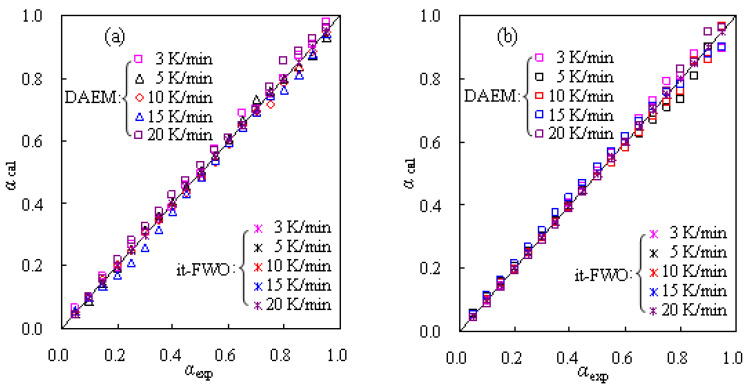
Calculated mass conversions for thermal decomposition of TBR wastes in air (**a**) and in N_2_ (**b**).

**Figure 10 materials-16-02328-f010:**
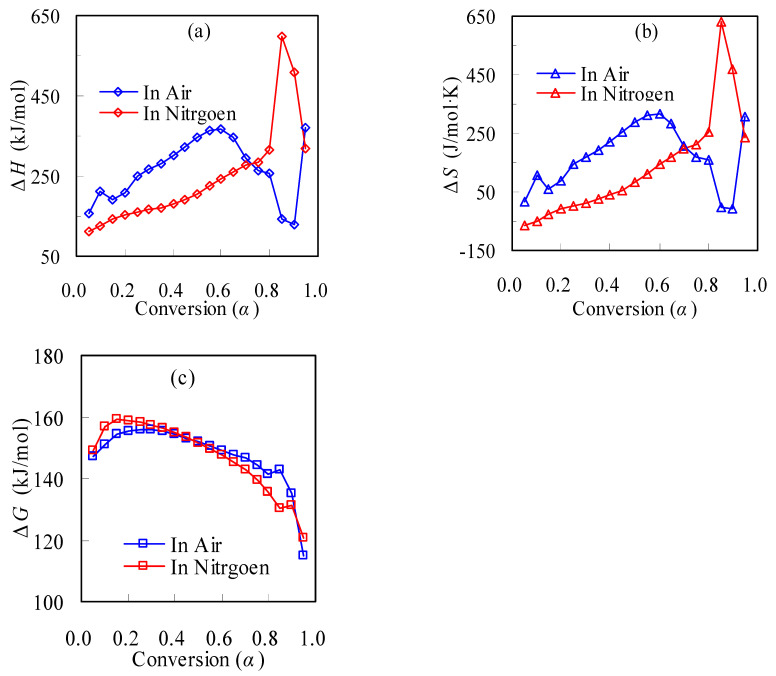
Conversion-dependent thermodynamic parameters for TBR wastes: (**a**) Δ*H*, (**b**) Δ*S* and (**c**) Δ*G*.

**Table 1 materials-16-02328-t001:** Some specific parameters for the thermal decomposition of TBR wastes.

Parameters	*β* (K/min)				
	3	5	10	15	20
In air					
*T*_5_ (K)	540.8	546.4	558.1	565.7	566.8
*T*_30_ (K)	632.2	639.0	647.0	649.7	656.6
*HRI* (K)	291.9	294.9	299.6	301.9	304.1
*T*_p_ (K)	648.9	654.2	661.2	670.0	673.8
*DrTGA*_p_ (min^−1^)	−0.030	−0.039	−0.082	−0.121	−0.175
Residual (wt.%)	34.96	35.26	35.28	35.79	36.32
In N_2_					
*T*_5_ (K)	533.3	548.2	560.0	568.7	570.7
*T*_30_ (K)	622.0	631.4	643.9	653.4	656.4
*HRI* (K)	287.4	293.1	299.1	303.6	304.8
*T*_p_ (K)	628.1	639.1	654.5	665.2	666.6
*DrTGA*_p_ (min^−1^)	−0.030	−0.054	−0.118	−0.193	−0.243
Residual (wt.%)	33.76	34.64	35.82	36.72	37.41

**Table 2 materials-16-02328-t002:** *CPI* parameters obtained at 10 K/min for thermal decomposition of TBR wastes.

Parameters	Zone			
	I	II	III	IV
In air				
*T*_i_ (K)	417.1	696.5	749.1	830.8
*T*_p_ (K)	661.2	713.2	779.8	869.7
*DrTGA*_p_ (min^−1^)	−0.082	−0.060	−0.022	−0.014
*DrTGA*_mean_ (min^−1^)	−0.023	−0.040	−0.013	−0.007
Δ*T* (K)	139.7	26.3	40.9	36.3
*CPI* (×10^11^ min^−2^·K^−3^)	5.01	18.11	1.24	0.35
*CPI*_total_ (×10^11^ min^−2^·K^−3^)	6.77			
In N_2_				
*T*_i_ (K)	401.8	589.2	749.2	791.4
*T*_p_ (K)	580.5	654.5	775.1	825.9
*DrTGA*_p_ (min^−1^)	−0.017	−0.118	−0.010	−0.012
*DrTGA*_mean_ (min^−1)^	−0.005	−0.049	−0.007	−0.005
Δ*T* (K)	93.8	80.0	21.1	102.2
*CPI* (×10^11^ min^−2^·K^−3^)	0.35	18.56	0.60	0.09
*CPI*_total_ (×10^11^ min^−2^·K^−3^)	14.51			

## Data Availability

The data presented in this study are available on request from the corresponding author.

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
