# Peer review of "Pyrolysis and Oxidative Thermal Decomposition Investigations of Tennis Ball Rubber Wastes through Kinetic and Thermodynamic Evaluations"

_materials, 2023, doi:10.3390/ma16062328_

Round 1

Reviewer 1 Report

The authors have written an interesting article and did a lot of work, however, I have a few comments:

1) Could author provide more information about tennis ball composition e.g. what kind of rubber they are made of?

2) Symbols (a) and (b) on Figures are in a poorly visible place. Please provide symbols e.g. at the top of the figure.

3) Figs. 2, 4, 6, 8– figures should be the same size.

Reviewer 2 Report

Presented manuscript reports the decomposition kinetics of tennis balls rubber wastes. In present form the methodology of the study does not fully meet the current standards in thermokinetics and need to be revised seriously. Most of issues with this manuscript have been discussed at length in reviews:

https://doi.org/10.3390/thermo2040029

https://doi.org/10.1016/j.tca.2022.179384

https://doi.org/10.1016/j.tca.2011.03.034 and so on. Please consult the literature and use it in your analysis

Briefly most important points:

- the authors used several isoconversional methods that are known to be of the different accuracy. Of three methods from this study, the Friedman method is known to be most accurate. The DAEM version used here is the most simple one and the most crude one too.

-15 mg can be generally regarded as too high sample mass for heating rates ~10 K/min, see the recommendations cited above 

- the meaning of the effective reaction orders ~5 can hardly be explained, in practice the reaction orders are less than 3

-the meaning of "thermodynamical" parameters derived from inprecise isoconversional methods for a very complex process are not clear for me either

- the kinetic data for a complex process shouldn't be reported with such an excessive accuracy (e.g. to 0.1 kJ/mol for Ea)

-the most interesting part, the mechanistic analysis of the kinetic findings from present study is missing

Round 2

Reviewer 2 Report

The authors address all my comments in revised manuscript and cover letter. Paper can be published